# Multilayer Dense Connections for Hierarchical Concept Prediction

## Abstract

Classification is a pivotal function for many computer vision tasks such as image recognition, object detection, scene segmentation. Multinomial logistic regression with a single final layer of dense connections has become the ubiquitous technique for CNN-based classification. While these classifiers project a mapping between the input and a set of output category classes, they do not typically yield a comprehensive description of the category. In particular, when a CNN based image classifier correctly identifies the image of a Chimpanzee, its output does not clarify that Chimpanzee is a member of Primate, Mammal, Chordate families and a living thing. We propose a multilayer dense connectivity for a CNN to simultaneously predict the category *and* its conceptual superclasses in hierarchical order. We experimentally demonstrate that our proposed dense connections, in conjunction with popular convolutional feature layers, can learn to predict the conceptual classes with minimal increase in network size while maintaining the categorical classification accuracy.

## 1 Introduction

Classification is a core concept for numerous computer vision tasks. Given the convolutional features, different architectures classify either the image itself (He et al., 2015; Szegedy et al., 2016), the region/bounding boxes for object detection (He et al., 2017; Liu et al., 2015), or, at the granular level, pixels for scene segmentation (Chen et al., 2018). Although early image recognition works employed multilayer classification layers (Krizhevsky et al., 2012; Simonyan & Zisserman, 2015), the more recent models have all been using single layer dense connection (He et al., 2016; Szegedy et al., 2016) or convolutions (Lin et al., 2017).

The vision community has invented a multitude of techniques to enhance the capacity of feature computation layers (Xie et al., 2017; Huang et al., 2017; Hu et al., 2018; Dai et al., 2017; Chollet, 2016; Tan & Le, 2019). But, the classification layer has mostly retained the form of a multinomial/softmax logistic regression performing a mapping from a set of inputs (images) to a set of categories/labels. As such, the final output of these networks do not furnish a comprehensive depiction about the input entity. In particular, when an existing CNN correctly identifies an image of an English Setter, it is not laid out in the output that it is an instance of a dog, or more extensively, a hunting dog, a domestic animal and a living thing. It is rational to assume that convolutional layers construct some internal representation of the conceptual superclasses, e.g., dog, animal etc., during training. We argue that, by appropriately harnessing such representation, one can retrieve a much broader description of the input image from a CNN than it is supplied by a single layer output.

Extensive information about most categories are freely available in repositories such as Word-Net (Fellbaum, 1998). WordNet provides the hierarchical organization of category classes (e.g., English Setter) and their conceptual superclasses (e.g., Hunting dog, Domestic animal, Living thing). However, a surprisingly limited number of CNNs utilize the concept hierarchy. The primary goal of almost all existing studies is to improve the category-wise classification performance by exploiting the conceptual relations, often via a separate tool.

Deng et al. (2014) and Ding et al. (2015) apply graphical models to capture the interdependence among concept labels to improve category classification accuracy. Other works either do not clarify the semantic meaning of the ancestor concepts (Yan et al., 2015) or impose a level of complexity in the additional tool (RNN) that is perhaps unnecessary (Hu et al., 2016). We have not found an

existing (deep learning) model that attempts to predict both the finer categories and the chain of ancestor concepts for an input image by a single network. The classical hedging method (Deng et al., 2012) computes either the finer labels or one of its superclasses *exclusively*, but not both simultaneously.

In this paper, we introduce a CNN to classify the category *and* the concept superclasses simultaneously. As illustrated in Figure 1, in order to classify any category class (e.g., English Setter), our model is constrained to also predict the ancestor superclasses (e.g., Hunting dog, Domestic animal, Living thing) in the same order as defined in a given ontology. We propose a configuration of multilayer dense connections to predict the category & concept superclasses as well as model their interrelations based on the ontology. We also propose a simple method to prune and rearrange the label hierarchy for efficient connectivity.

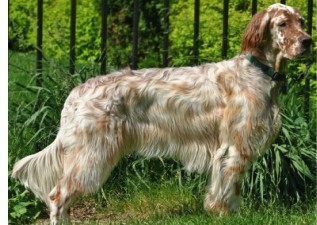

Existing methods: English Setter

*Proposed prediction: Living thing ➜ Domestic ➜ Hunting Dog ➜ English Setter.*

Figure 1: The goal of the proposed algorithm. In contrast to the existing methods, our proposed CNN architecture predicts the chain of superclass concepts as well as the finer category.

Capturing the hierarchical relationship within the CNN architecture itself enables us to train the model end-to-end (as opposed to attaching a separate tool) by applying existing optimization strategies for training deep networks[1]. We experimentally demonstrate that one can train the proposed architecture using standard optimization protocols to predict the concept classes with two popular CNN backbones: ResNet and InceptionV4, while maintaining their category-wise accuracy. The proposed multilayer connection is shown to further refine the learned representations of these backbone CNNs to yield better concept and category classification than 1) multinomial logistic regression, and, 2) other existing works (that apply separate mechanisms) on standard datasets and challenging images.

Predicting coarser superclasses in addition to finer level categories improves interpretability of the classifier performance. Even if an eagle is misclassified as a parrot, the capability of inferring that it is a bird, and not an artifact (e.g., drone), may be beneficial in some applications (e.g., surveillance). More importantly, an object detector can enhance its capability on unseen categories by adopting the proposed classification scheme (as demonstrated in Section 4.4). For example, a movie/TV violence recognition/detection tool can recognize an equipment as a 'weapon' concept class even if that particular weapon category was not in the training set. In visual question answering (VQA), encoding concept classes would expand the scope of query terms by allowing broader description ('how many vehicles are present' in addition to 'how many buses', 'how many trucks' etc.; see Cao et al. (2018); Wang et al. (2016)). In Appendix G, we point out how our architecture can be extended to object detectors to compute the concept classes. In addition, we allude to the potential applications of our model to capture label structures different from concept graph, e.g., spatial or compositional dependence.

## 2 RELEVANT WORKS

Use of hierarchical classifiers can be traced back to the early works of Torralba et al. (2004); Wu et al. (2004); Fergus et al. (2010) that shared features for improved classification. Some studies claimed a hierarchical organization of categories resembles how human cognitive system stores knowledge (Zhao et al., 2011) while others experimentally showed a correspondence between structure of semantic hierarchy and visual confusion between categories (Deng et al., 2010). Bengio et al. (2010); Deng et al. (2011) learn a label tree for efficient inference with low theoretical complexity and also suggest a label hierarchy is beneficial for datasets with tens of thousands of categories.

Deng et al. (2012) aim to predict either a coarse level concept or a fine level category (but not both at the same time) given an initial classical classifier. Provided an initial classifier output, this method determines category or the coarse concept node (exclusively) with max reward based on aggregated probabilities in a label hierarchy. The reported results suggest the prediction of superclasses comes at the expense of the fine level category classification failure. For CNN based classification, Deng et al. (2014) modeled the relationships such as subsumption, overlap and exclusion among the categories

---

[1]One can also envision learning the multilayer connectivity structure from data by architecture learning techniques (Zoph et al., 2017; Pham et al., 2018).

via a CRF. Although the CRF parameter can be trained via gradient descent, the inference required a separate computation of message passing. The work of Ding et al. (2015) extended this model by utilizing probabilistic label relationships.

The HDCNN framework (Yan et al., 2015) groups the finer classes into coarse concept using the category labels. The framework comprises two modules for coarse and fine categories where the coarse prediction modulates the layers for finer classification. However, this work does not describe the coarser concept classes in order to analyze whether they, or their descendent categories, have any semantic meaning (see Appendix H). Furthermore, the overlap among concept class descendants precludes a mechanism to predict the conceptual superclasses. Hu et al. (2016) present a structured inference model for hierarchical representation where the concepts representing scene attributes are predicted as indicator vectors of different length. A bidirectional message passing, inspired by the bidirectional recurrent networks, establishes the relations among different levels of concepts. The model leads to a large number of inter and intra-layer label interactions some which needed to be manually hard-coded to 0.

Appendix A describes a few other studies that proposed to incorporate structural prediction techniques in their design (Guo et al., 2017; Liang, 2019). Unlike us, the primary objective of almost all the aforementioned papers is to improve the category prediction performance by utilizing the superclass hierarchy as an auxiliary source of information or as intermediate result. With the exception of Deng et al. (2012); Hu et al. (2016), none of these studies were designed for, and demonstrate their effectiveness in, concept prediction. The algorithms of Deng et al. (2014); Guo et al. (2017) require substantial modifications to be able to classify the concept classes. Furthermore, in contrast to ours, most of these studies use a separate technique/tool for modeling the conceptual relations that need to be trained or applied separately with different mechanisms.

It is important to distinguish our work from the hyperbolic embedding studies (Nickel & Kiela, 2017; Khrulkov et al., 2020). Khrulkov et al. (2020) attempt to compute an embedding that respects ancestral hierarchy in the hyperbolic space – which implies that, ideally, a more generic image would be closer to the center than the specific ones. However, the paper does not describe – and, it is neither obvious nor straightforward – how to determine the concept classes from these embedded points. These studies, therefore, are not suitable for a comparison with our method for concurrent prediction of category and concept classes.

## 3 PROPOSED METHOD

Given an input image, the goal of our proposed method is to determine its category (leaf node in the hierarchy) *and* a list of its concept superclasses (i.e., ancestors in the ontology). As an example, for an image of a Chimpanzee, the proposed algorithm produces predictions for 1) the category Chimpanzee, and 2) an ordered list of ancestor concepts: Living thing → Chordate → Mammal → Primate → Chimpanzee. We illustrate (and experiment with) the proposed model for image classification in this paper.

Our CNN architecture is designed to encompass the chain of relationships among the category and the predecessor concepts in the dense layers. We utilize an existing label hierarchy/ontology to guide the design of the dense layers, but do not use the hierarchy in prediction. In order to maximize the information within an ontology and to reduce the number of variables in the dense layers, we condense the original label hierarchy as explained in Section 3.1. In our design of multilayer dense connections, each concept is associated with a set of hidden nodes. These hidden nodes are in turn connected to those of its children (both category and concept) and the output prediction nodes. Section 3.2 elaborates these connections and the loss functions to train the network.

### 3.1 CONDENSED CONCEPT HIERARCHY

In general, a concept class decomposes to multiple sub-concepts in an ontology, e.g., ImageNet12 (Deng et al., 2009) subset of WordNet (Fellbaum, 1998). However, the lineage of a parent to a single child (e.g., Entity → Physical Entity → Abstraction ) is redundant and does not provide much information. Similarly, parent concepts with highly imbalanced distribution of descendants are not informative as well. Modeling the redundant and uninformative concepts will increase the network size with no information gain.

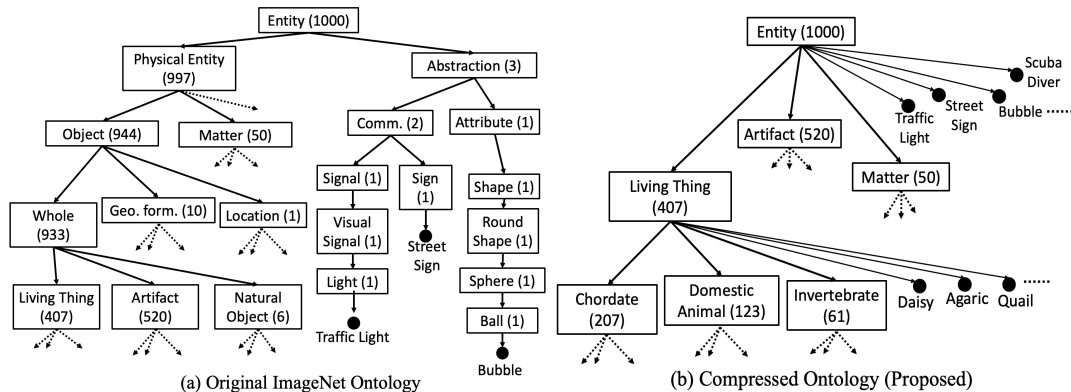

(a) Original ImageNet Ontology          (b) Compressed Ontology (Proposed)

Figure 2: Partial view of the original (left) and condensed (right) label hierarchies. Concepts are enclosed in rectangular boxes, with number of all descendeants in parentheses. All the parent concepts of the categories Traffic Light, Street Sign and Bubble in the original ontology are absorbed to Entity in the compressed ontology by removing the redundant single parent-child connections and excluding nodes with descendant count $< \delta$. Similarly, the ancestors (Physical Entity, Object, Whole) of Living Thing, Artifact are all collapsed to Entity because $\tau\%$ descendants of each parent in the ancestry were also descendants of its child.

We reorganize the given ontology to reduce such redundancy. We assume the hierarchy to be a directed acyclic graph (DAG) and perform a depth first search (DFS) traversal on it. During the traversal, we first prune the label hierarchy based on the distribution of descendants of a concept node. Let $\eta_\gamma$ denote the number of all descendants of a concept indexed by $\gamma$. Any child node $\gamma_{\text{ch}} \in \texttt{Children}(\gamma)$ with $\frac{\eta_{\gamma_{\text{ch}}}}{\eta_\gamma} \geq \tau$, i.e., the overlap between descendants $\gamma$ and $\gamma_{\text{ch}}$ is more than $\tau$, is absorbed by the parent $\gamma$. This process is applied recursively to yield a balanced distribution of descendants of any concept in the resulting hierarchy. In addition, we remove any concept $\gamma$ in the structure with a descendant count $\eta_\gamma < \delta$ and append the children set $\texttt{Children}(\gamma)$ to those of its parent $\gamma_{\text{pa}}$. Conceptually, it is not worth modeling a concept node with only a few descendants.

We depict the differences between the original and modified ontologies in Figure 2. The distributions of the descendants for children concepts are more balanced in the compressed version (right) than that in the original version (left). Our strategy removes all the redundant single parent-child connections. As the network connections are dependent on the concept hierarchy, this reduction of nodes and relationships in the ontologies are crucial for our method. It is worth noting that the proposed modification added direct concept-category relations in the middle layers of the hierarchy.

Executing a DFS on a DAG ontology may lead to a few ambiguous grouping of few concepts and categories as Deng et al. (2014) pointed out. We adopted DFS for simplicity to compute the compressed graph, which can be replaced by an unambiguous one whenever it is available; the methods of Deng et al. (2011); Bengio et al. (2010) can also be applied to generate the ontology.

### 3.2 Network Architecture

Our proposed algorithm aims to model the abridged label hierarchy with dense connections. As Figure 2 suggests, there are multiple kinds of dense connectivities in our proposed classification layer. Each concept in the hierarchy corresponds to one set of hidden nodes that essentially represent the concept. These hidden nodes are connected to those representing its children, if any. For example, if Mammal, Bird and Reptile are the descendant concept of Chordate, there will be all to all connections from the hidden nodes representing Chordate to those accounting for Mammal, Bird and Reptile. As such, the computation of the hidden representation of a child concept is conditioned upon that of its parent.

Given the representation captured in the hidden nodes, two types of output prediction nodes detect the presence of the concept itself and any children category in the input. An additional type of connectivity explicitly constrains the concept and category predictions to follow the hierarchical organization of the ontology. We illustrate each of these connections below.

**Modeling Concepts and Categories:** Let us denote by $z^\gamma$ and $\mathbf{h}^\gamma$ the output prediction variable and the set of hidden nodes associated with the concept $\gamma$. The terms node and variables are used interchangeably in the description of our model. Let concept $\gamma_{\text{ch}}$ and category $j$ both be children of concept $\gamma$ in the hierarchy and $\mathbf{h}^{\gamma_{\text{ch}}}$ and $x_j$ denote the hidden and the output prediction variables for them respectively. The proposed model computes the output prediction $z^\gamma$ and initial values $\tilde{\mathbf{h}}^{\gamma_{\text{ch}}}$,

$\tilde{x}_j$ for quantities of the children concept and categories using the following dense connections.

$$z^\gamma = \phi\Big( \sum_i u_i^\gamma h_i^\gamma + b_z^\gamma \Big); \quad \tilde{x}_j = \psi\Big( \sum_i v_{i,j}^\gamma \, h_i^\gamma + b_j^\gamma \Big); \quad \tilde{h}_{i,i'}^{\gamma_{\mathrm{ch}}} = \omega\Big( \sum_i w_{i,i'}^\gamma \, h_i^\gamma + b_{i'}^\gamma \Big) \qquad (1)$$

In these equations, $u, v, w$ and $b$ are the weights/biases of the dense connectivity and $h_i$ corresponds to the $i$-th value of $\mathbf{h}$. The activation functions $\phi, \psi, \omega$ utilized for these different quantities are $\phi = \mathrm{Sigmoid}$, $\psi = \mathrm{Identity}$, $\omega = \mathrm{ReLU}$. In essence, $\mathbf{h}^\gamma$ encodes an internal description for concept $\gamma$ and $z^\gamma$ predicts the presence of it in the input image. The connections between $\mathbf{h}^\gamma$ and $\mathbf{h}^{\gamma_{\mathrm{ch}}}$ of its child concept enforces the descriptions for $\gamma_{\mathrm{ch}}$ to be derived from, and therefore dependent on, that of its parent.

In our design, the number $d^\gamma = |\mathbf{h}^\gamma|$ of nodes representing a concept $\gamma > 0$ is directly proportional to the total number of descendants $\eta_\gamma$ of $\gamma$. We have used $d^\gamma = \mu \eta_\gamma$ for this study with $\mu = 2$. The flattened output of the final feature layer of an existing network architecture (e.g., ResNet-50 or InceptionV4 etc.) is utilized to populate $\mathbf{h}^0$ and its size depends on the particular architecture used. We do not predict the root concept $z^0$ of the hierarchy (e.g., Entity for ImageNet12) since all categories descend from it.

**Concept Category Label Constraints:** The values for category prediction $x_j$ and hidden nodes for child concept $\mathbf{h}^{\gamma_{\mathrm{ch}}}$ are calculated by multiplying initial values of these quantities with the concept prediction $z^\gamma$.

$$x_j = \tilde{x}_j * z^\gamma; \quad h_{i'}^{\gamma_{\mathrm{ch}}} = \tilde{h}_{i'}^{\gamma_{\mathrm{ch}}} * z^\gamma \qquad (2)$$

Note that, the Sigmoid activation constrains the value of $z^\gamma$ to be $z^\gamma \in [0, 1]$. In effect, the node $z^\gamma$ plays an excitatory or inhibitory role based on the predicted value of the concept $\gamma$. This constraint enforces that the nodes representing any child of concept $\gamma$, whether it is a category $(x_j)$ or another downstream sub-concept $(\mathbf{h}^{\gamma_{\mathrm{ch}}})$, be activated only if the concept itself is correctly predicted.

The category predictions for an input image are computed by applying Softmax activation over all category nodes $\{x_1, x_2, \ldots, x_N\}$, where $N$ is the total number of categories. The predictions for $M$ concepts are given by the collection of the variables $\{z^1, z^2, \ldots, z^M\}$. The hierarchical relationship among the variables $\{z^1, z^2, \ldots, z^M\}$ are enforced by construction. Observe that, while an image can be classified to only one (e.g., Chimpanzee) of the $N$ categories, multiple concepts (e.g., Primate, Mammal, Chordate) at different levels of the hierarchy can be set to 1.

Figure 3 clarifies the proposed dense arrangement between the hidden nodes $\mathbf{h}^\gamma$ and its children concept nodes as well as the prediction outputs. The hidden nodes $\mathbf{h}^\gamma$ (shown in empty circles) are connected to those of its $Q$ children concepts and $B$ category output variables (solid circles) to compute the initial quantities $\{\tilde{\mathbf{h}}^{\gamma_{\mathrm{ch}_1}}, \ldots, \tilde{\mathbf{h}}^{\gamma_{\mathrm{ch}_Q}}\}$ and $\{\tilde{x}_{j_1}, \ldots, \tilde{x}_{j_B}\}$ respectively. The concept prediction $z^\gamma$ (solid square) is computed by another dense connections which modulates the final values of $\{\mathbf{h}^{\gamma_{\mathrm{ch}_1}}, \ldots, \mathbf{h}^{\gamma_{\mathrm{ch}_Q}}\}$ and $\{x_{j_1}, \ldots, x_{j_B}\}$ for the concept and category variables respectively via multiplication.

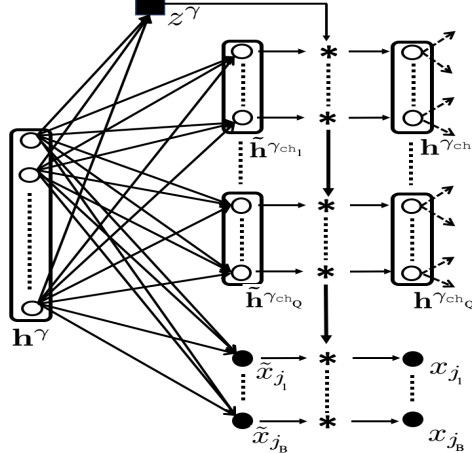

Figure 3: Schematic view of proposed dense connections. The solid square and circle nodes correspond to the concept and category prediction node respectively, whereas the empty circles depicts the hidden nodes. We assume the concept $\gamma$ has $Q$ concepts and $B$ categories as children.

**Number of Variables:** The total number of weights in proposed multilayer dense connection with $\alpha$-way decomposition is $\leq \mu d^0\big( N + \rho + \frac{1}{(1-\alpha)} \big)$ where $\rho$ is the height of the hierarchy and $\mu$ is the fixed multiplier used to set the number of hidden nodes for concept $\gamma$ (see Appendix B). In comparison, conventional single layer CNN classifiers consist of $d^0 N$ variables.

**Loss Functions:** The proposed method minimizes two different losses for the two types of output nodes. For the category predictions, we minimize a cross-entropy loss $L_{CE}(x)$ computed over the $N$ category labels and the network outputs $x_j$, $j = 1, \ldots, N$. The concept classification loss is defined as the MSE between the concept prediction

variables $\mathbf{z}$ and a binary indicator quantity $\boldsymbol{\chi}$ : $L_{CON}(z) = \frac{1}{M} \sum_{\gamma=1}^{M} (z^\gamma - \chi^\gamma)^2$, where $\chi^\gamma = 1$ if $\gamma$ is an ancestor concept of the input category and $\chi^\gamma = 0$ otherwise for $\gamma = 0, 1, \ldots, M$.

Although we use MSE for learning $\mathbf{z}$ in this work, any loss for multilabel classification, e.g., binary cross-entropy for each $z^\gamma$, may work as well. The proposed method minimizes the combined loss $L_{CE} + \lambda L_{CON}$ with the balancing weight $\lambda$ fixed to $\lambda = 5$ in all our experiments. Note that, while the error for any category is backpropagated through its predecessor concepts due to the dependence imposed by construction, one needs to ensure that other concepts – that are not related to the category – to remain 0 in the $\mathbf{z}$. This is exactly the constraint enforced by $L_{CON}(z)$.

## 4 EXPERIMENTS & RESULTS

### 4.1 EXPERIEMNTAL SETUP

We utilize the ontology provided by the ImageNet12 dataset Deng et al. (2009) to design our dense layers. All the labels of ImageNet12 between $[1, 1000]$ correspond to the $N = 1000$ category classes and labels $> 1000$ are assigned to the 860 concept superclasses. In all our experiments, we fixed the two quantities for compressing the concept hierarchy (Section 3.1) to be $\tau = 90\%$, $\delta = 20$. After compressing the label hierarchy using the methods described in Section 3.1, there are 40 concept labels left in the hierarchy which has a height of $\rho = 7$.

The proposed architecture has been trained on the ImageNet12 training dataset and tested on multiple datasets as described in next sections. We used two popular CNN architectures for feature computation layers: ResNet-50 (He et al., 2015; 2016) and Inception V4 (Szegedy et al., 2015; 2016)[2]. Adding the proposed multilayer dense connections increased the total number of variables of the ResNet-50 model by a factor of 1.182 ($\frac{30.29}{25.61}$M). For the InceptionV4 model, the increase is 1.059 ($\frac{46.89}{44.24}$M). This suggests that the increase induced by our proposed model is far lower than the analytical estimate (Section 3.2) in practice and is tolerable with respect to overall network size.

**Inference:** During inference of the proposed network, we select the category with largest softmax probability for category classification as usual. The parent concepts are classified by thresholding $\mathbf{z}$ values. We also set any $z^\gamma = 0$ if the variable for its parent $z^{\gamma_{pa}} = 0$ (i.e., lower than a confidence threshold). If more than one child of any concept is detected, we select the one with the highest confidence among them to compute the concept chain.

**Evaluation measures:** We report the single crop top-1 accuracy results $Acc_{CAT}$ for categories when they are available. For concept inference, we report two measures based on hierarchical precision and recall (Costa et al., 2007), which computes the precision an recall between the predicted and true chain of concepts : 1) the mean hierarchical precision (mhP) and recall (mhR) over all images of test set, and, 2) percentage of images $Acc_{CON}$ that were predicted with 100% hP and hR. The combined accuracy, $Acc_{COMB}$, denotes the percentage of images with both $Acc_{CAT} = Acc_{CON} = 1$.

**Baselines:** As the first baseline, we build a CNN classifier for $N + M$ classes with a single flat dense layer for each feature computation architecture. These baselines are trained with the same loss functions of the their multilayer counterparts.

We utilize a modified version of the hedging method of Deng et al. (2012) as the second baseline. Our modified version uses ReasNet-50 and Inception V4 based classifiers instead of classical versions. We observed the optimal strategy to compute category and concept chain from the output of the hedge model (which is either category or max reward concept) is to use the category prediction as is and use the concept with maximum reward, along with all its ancestors, as the predicted concept chain. The hedge model utilizes the same underlying hierarchy as ours and we read off the prediction for concepts that overlap with our condensed hierarchy.

### 4.2 IMAGENET12 DATASET RESULTS

We first demonstrate the performance of two varants of the proposed method that were trained on ImageNet12 training set and tested on its validation set. For one variant of our proposed CNN models, **MD-RN** with ResNet-50 and **MD-IC** with Inception V4 backbones, only the weights of

---

[2]pretrained versions downloaded from `https://github.com/tensorflow/models/tree/master/research/slim`.

Table 1: Accuracy comparison on ImageNet12 val set (single crop top-1). The proposed models achieve significantly higher $Acc_{\text{CON}}$, $Acc_{\text{COMB}}$ than the baselines.

| Method | $Acc_{\text{CAT}}$ | $Acc_{\text{CON}}$ | $Acc_{\text{COMB}}$ | mhP | mhR |
|---|---|---|---|---|---|
| HG-RN (ResNet+Deng et al. (2012)) | 68.97 | 82.9 | 66.15 | 93.28 | 94.57 |
| BL-RN | 76.26 | 62.66 | 53.01 | 92.36 | 89.48 |
| BL-RN-FT | 76.0 | 69.04 | 58.89 | 93.91 | 91.06 |
| MD-RN(ours) | 75.94 | 80.25 | 69.05 | 94.22 | 93.68 |
| MD-RN-FT(ours) | 75.91 | 82.36 | **70.45** | 93.3 | 94.75 |
| HG-IC (Inception+Deng et al. (2012)) | 71.49 | 84.28 | 68.87 | 93.75 | 95.07 |
| BL-IC | 80.11 | 72.68 | 63.56 | 95.03 | 91.51 |
| MD-IC(ours) | 80.02 | 88.18 | **77.76** | 95.46 | 96.1 |

the multilayer dense connections were trained; the weights of the feature layers were kept at their original pretrained values. The other variant **MD-RN-FT** performs finetuning, i.e., it initiates the weights in the feature layer from the pretrained ResNet-50 and trains all variables in the network. The corresponding flat dense connection baselines are referred to as BL-RN, BL-IC, BL-RN-FT respectively. The hedging model of Deng et al. (2012) is dubbed HG-RN and HG-IC that utilizes a ResNet-50 and Inception V4 initial classifiers respectively.

**Training:** We trained MD-RN, MD-RN-FT and MD-IC following standard training procedures of Szegedy et al. (2016); Chollet (2016). The details of training parameters and techniques are described in Appendix C for space limitation. The initial classifiers for HG-RN and HG-IC were trained from scratch on 93% of training examples for each class in ImageNet 2012 training set (achieving 74% and 76% category accuracy on validation set respectively). The remaining 7% examples were used to estimate the dual variable maximizing the reward for multiple accuracy guarantees. The results reported here correspond to the dual variable, accuracy guarantee with best overall $Acc_{\text{COMB}}$.

**Evaluation:** Table 1 reports the accuracies for all the aforementioned models. Table 1 demonstrates that the MD-RN, MD-RN-FT and MD-IC networks with proposed multilayer dense connections achieved a comparable category accuracy $ACC_{\text{CAT}}$ of the flat baselines BL-RN, BL-RN-FT and BL-IC (which are similar to the published results). The category accuracy of HG-RN, HG-IC are significantly lower than the aforementioned models – a phenomenon also observed by Deng et al. (2012). This is due to the inherent design of the hedge model to only predict concept classes (and not the leaf categories) for some examples.

Table 1 also suggests that an unconstrained flat dense connection is insufficient for learning the superclass sequence. The proposed models attains $> 10\%$ higher $Acc_{\text{CON}}$ and even larger combined $Acc_{\text{COMB}}$ than the flat baselines BL-RN, BL-RN-FT, BL-IC. The baseline methods can either learn to classify the finer categories or the concept chain correctly but the proposed CNN architecture can attain the ability to do both simultaneously with high accuracy. The proposed models MD-RN, MD-RN-FT and MD-IC achieves a significantly higher overall accuracy $Acc_{\text{COMB}}$ than the hedging models HG-RN and HG-IC. The inception based model MD-IC is also more accurate than HG-IC for concept prediction – indeed, MD-IC leads to the best overall accuracy $Acc_{\text{COMB}}$ among all models.

The flat baseline methods BL-RN, BL-RN-FT and BL-IC attempts to compute the concept classes independently using the same single layer dense connections that are utilized for category classification. Observe that the concept accuracies $Acc_{\text{CON}}$ of BL-RN, BL-RN-FT and BL-IC are approximately $7 \sim 14\%$ lower than category accuracies $Acc_{\text{CAT}}$. This suggests that the feature representations learned (in the last convolutional layer) by ResNet-50 and Inception V4 alone are not as informative for concept prediction as they are for category classification. The proposed multilayer design combines these representations to yield a refined representation encoded in the hidden variables $h^\gamma$ to improve the concept prediction accuracy by $> 13\%$.

The comparison implies that the proposed top-down hierarchical constraints on category classification to preserve the ancestor concept chain is superior to the hedging technique of concept prediction distilled from the aggregated probabilities accumulated bottom-up from category confidence. As we will show in next section, a top-down constrained prediction is more robust to the challenging examples because it does not rely primarily on the category classification probabilities.

**Analysis:** We provide an in-depth analysis of the failure cases of concept prediction, compared the distribution of category classifications and training times of different variants of the proposed models and baslines in Appendix D. Appendix E discusses the ablation studies with the network structure and $\lambda$ values in loss function.

Table 2: Accuracy comparison on naturally adversarial images (single crop top -1). The proposed models achieve significantly higher $Acc_{CON}, Acc_{COMB}$ than the baselines.

| Method | $Acc_{CAT}$ | $Acc_{CON}$ | $Acc_{COMB}$ | mhP | mhR |
|---|---|---|---|---|---|
| HG-RN (ResNet+Deng et al. (2012)) | 0.37 | 14.36 | 0.3 | 57.08 | 55.99 |
| BL-RN-FT | 0.86 | 11.52 | 0.26 | 59.12 | 54.33 |
| MD-RN-FT (ours) | 1.06 | **16.13** | 0.77 | 57.6 | 58.0 |
| HG-IC (Inception+Deng et al. (2012)) | 3.18 | 19.76 | 2.8 | 59.73 | 58.77 |
| BL-IC | 7.54 | 16.62 | 3.56 | 61.78 | 58.07 |
| MD-IC (ours) | 7.1 | **23.5** | 6.1 | 63.79 | 61.46 |

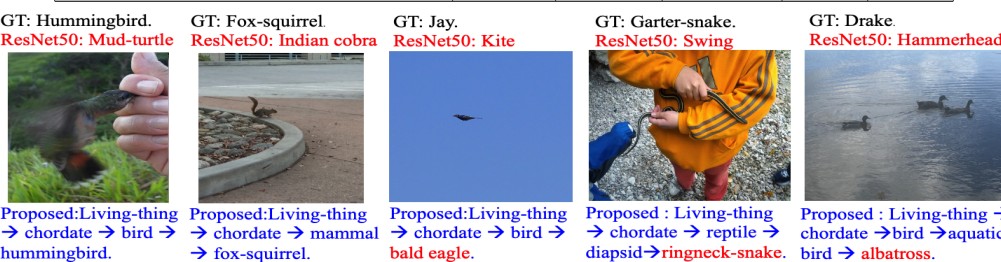

GT: Hummingbird.
ResNet50: Mud-turtle

GT: Fox-squirrel.
ResNet50: Indian cobra

GT: Jay.
ResNet50: Kite

GT: Garter-snake.
ResNet50: Swing

GT: Drake.
ResNet50: Hammerhead

Proposed:Living-thing → chordate → bird → hummingbird.

Proposed:Living-thing → chordate → mammal → fox-squirrel.

Proposed:Living-thing → chordate → bird → bald eagle.

Proposed : Living-thing → chordate → reptile → diapsid→ringneck-snake.

Proposed : Living-thing → chordate →bird →aquatic-bird → albatross.

Figure 4: Qualitative comparison between the proposed and the pretrained ResNet50 models on naturally adversarial images (Hendrycks et al., 2019).

### 4.3 PERFORMANCE ON NATURALLY ADVERSARIAL IMAGES

Our next experiment demonstrates the robustness of the proposed network on challenging images. We use the images of naturally adversarial image dataset (Hendrycks et al., 2019) that the popular CNN architectures (e.g. ResNet) trained on standard datasets classify to widely different semantic categories. Results in this section examine whether or not algorithms (e.g., Deng et al. (2012)) that compute marginal probabilities through bottom up aggregation are less effective for determining the ancestor concepts of challenging images. There are 7500 images in this dataset that belong to 200 categories. The ResNet-50 model we downloaded from the tensorflow website (pretrained on full ImageNet 2012) could classify only 0.73% of the images. Since these 200 categories are a subset of the $N = 1000$ categories of ImageNet 2012 dataset, one can use the same concept structure for hierarchical classification.

In Table 2 the proposed MD-RN-FT and MD-IC model accuracies are compared with those of their flat dense layer counterparts BL-RN-FT and BL-IC as well as those of the hedging model (Deng et al., 2012) with corresponding initial classifiers. The proposed models MD-RN-FT and MD-IC were able to correctly classify the superclass sequences of more images than the baseline models. With the exception of BL-IC, their categorical accuracies $ACC_{CAT}$ are also higher than the baselines.

We provide a qualitative comparison in Figure 4 between the proposed MD-RN-FT method and the pretrained ResNet-50 output. One can observe that, without the constraints on predicting the concept chain correctly, the ResNet-50 can classify the images to semantically very different categories. Aggregating these category-wise confidences will consequently lower the $ACC_{CON}$ of the hedging model compared to those of the proposed model. The proposed MD-RN-FT, on the other hand, determines the ancestor classes correctly even when the leaf level category is misclassified.

### 4.4 PASCAL VOC 12 CONCEPT PREDICTION

In this section, we test the generalization of the knowledge learned by the proposed architecture. Ideally, the proposed network should be able to extrapolate its understanding of the concept super-classes learned from one category set to previously unseen categories. That is, after learning that a zebra is a mammal, it should be able to identify a horse (of any color) as a mammal too. To test generalization capability, we test classifiers learned from the ImageNet12 dataset on the trainval split of PASCAL VOC dataset (Everingham et al., 2010). Note that, there is no one to one correspondence between the categories of ImageNet and VOC[3]. Each of the VOC 12 categories were assigned to one of the concepts in the condensed label hierarchy of ImageNet dataset.

In Table 3, we show the average concept prediction accuracies of the different variants of the proposed model and the corresponding flat dense layer baselines over 16 VOC categories. Since there are no categories to predict, the second baseline predicts the concept classes by thresholding the aggregated probabilities as computed in Deng et al. (2012). These are referred to as PrAggRN and

---

[3]See http://image-net.org/challenges/LSVRC/2012/analysis.

Table 3: Average concept accuracy comparison on PASCAL VOC 12 trainval subset. The proposed method achieves significantly higher $Acc_{\text{CON}}$ than the baseline for both architectures.

| | PrAggRN | BL-RN-FT | MD-RN-FT(ours) | PrAggIC | BL-IC | MD-IC(ours) |
|---|---|---|---|---|---|---|
| $Acc_{\text{CON}}$ | 46.2 | 38.06 | **48.84** | 48.32 | 40.3 | **50.38** |

Table 4: Accuracy comparison on AwA2 dataset for simultaneous category, concept, attribute classification.

| Method | $Acc_{\text{CAT}}$ | IoU Concept | IoU Attribute |
|---|---|---|---|
| Hu et al. (2016) | $79.36 \pm 0.43$ | $84.47 \pm 0.38$ | $86.92 \pm 0.18$ |
| BL-RN | $92.48 \pm 0.05$ | $93.66 \pm 0.42$ | $89.08 \pm 0.01$ |
| MD-RN (ours) | $92.16 \pm 0.17$ | $\mathbf{97.01 \pm 0.05}$ | $\mathbf{94.1 \pm 0.37}$ |

PrAggIC when computed from the initial ResNet-50 and Inception V4 classifiers respectively. As the table shows, the proposed variants MD-RN-FT, MD-RN-SC and MD-IC achieved a higher average concept prediction accuracy over all classes than those of the baselines. Appendix F details the performances for all classes and analyzes the results on this dataset.

## 4.5 AWA2 DATASET RESULTS

Our next experiment applies the proposed method on the Animals with Attributes (AwA2) dataset (Lampert et al., 2014) utilized in multiple relevant studies (Deng et al., 2014; Hu et al., 2016). AwA2 dataset provides the attribute as well as the category labels for 37322 images of 50 classes. For concept classes, we use the same 28 superclasses and follow the same hierarchical relations used in Hu et al. (2016). Note that, this hierarchy is slightly modified from the ImageNet12 ontology (and therefore different from the compressed ontology we computed).

Following Hu et al. (2016), we use a 60-40% split of the images for train/test and report the average±std dev of the multiclass accuracy $Acc_{\text{CAT}}$ for category and the IoU acc for concept and attribute labels from 3 trials. Table 4 compares the performances of the proposed MD-RN with that of (Hu et al., 2016) (values taken from the paper). To determine the attributes, we extend our proposed architecture by adding a dense connection between concept and category prediction variables to the 85 attribute classes. We also compare the performance of the flat baseline BL-RN that predicts the attributes classes independently in addition to animal and concept labels.

Hu et al. (2016) uses a weaker CNN base ((Krizhevsky et al., 2012)) than ours, which is perhaps a factor leading to its inferior performance. The comparison suggests that, given CNN backbone with enhanced capacity, one can attain highly correct category, concept and attribute classification from our multilayer dense connection without relying on an additional complex tool, e.g., RNN. However, the lower IoU concept, IoU attribute of the flat baseline method attest that the improved performance of MD-RN does not arise solely from the ResNet-50 features and strengthen our claim that the proposed mulitlayer connectivity enhances the feature representation to improve accuracy. It is interesting to observe a correlation between concept and attribute accuracies: determining the concepts correctly leads to a better prediction of attributes. Appendix I examines the robustness of MD-RN trained on AwA2 on ImageNet 12 images from categories overlapping with AwA2.

## 5 DISCUSSION

In this paper, we have demonstrated the advantages of multilayer dense connections for classifying both the category and the ordered chain of ancestor concept classes with two popular architectures in different settings. The experimental results show that the proposed dense layers can equip the existing CNN classifiers with additional capability of learning the coarse concepts without sacrificing the category-wise accuracy. Attaining this capability of determining the ancestor concept classes is relatively easy – one can only modify the final classification layers and train using the same optimization technique, for some cases without training the full network.

Our experiments demonstrate the advantage of the proposed architecture over a) multinomial regression that exploits the learned feature representation of pretrained CNNs; b) methods utilizing marginal probabilities accumulated in a bottom up fashion from CNN output given an offline hierarchy (Deng et al., 2012); and c) an algorithm that employs RNN for superclass prediction. Our top-down constraints enforces the category classification to conform to the semantically meaningful lineage of ancestor classes. This property not only improves the superclass computation for challenging images but also allows zero-shot concept classification for unseen categories. We believe this study will encourage researchers in computer vision community to adopt and improve our multilayer dense connection to gather more descriptive output from CNNs.

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

## A    APPENDIX: OTHER RELEVANT WORKS

Guo et al. (2017) attempted to classify the coarse labels or the conceptual superclasses of categories by augmenting an RNN to CNN output. In addition to increased complexity imposed by the RNN, it is not clear how the hierarchy among labels was generated and how the hierarchy would scale up with increasing number of categories. Liang (2019) employed graph based reinforcement learning to learn features of a set of modules, each of which corresponds to a concept. Once the network search is completed, the activated module outputs are passed through prediction layers for generating final output. The algorithm demonstrated promising performance for classifying scene contexts for semantic segmentation.

Commercial vision solutions such as Amazon Rekognition (https://aws.amazon.com/blogs/aws/amazon-rekognition-image-detection-and-recognition-powered-by-deep-learning/) or Google Vision (https://cloud.google.com/vision) seem to produce outputs that resemble concept classes. Due to the proprietary nature of these solutions, it is not possible for us to confirm whether or not they indeed predict concept classes and their relationships. However, based on the description provided in (https://aws.amazon.com/blogs/aws/amazon-rekognition-image-detection-and-recognition-powered-by-deep-learning/) and scrutiny of results, we speculate that Amazon rekognition does not use any hierarchy and probably uses a look-up from hard-coded ( undisclosed) relations strategy.

## B    APPENDIX: NUMBER OF VARIABLES IN MULTILAYER DENSE CONNECTIONS

This section quantifies the increase in the number of weights in the dense layers induced by our multilayer computations. The CNN classifiers typically consists of $d^0 N$ connections in dense layer, where $d^0$ and $N$ are the size of the last feature layer and number of category classes respectively. In proposed multilayer dense connections with a balanced $\alpha$-way decomposition of the concepts, the total number of weights is $\leq \mu d^0 \left( N + \rho + \frac{1}{(1-\alpha)} \right)$ where $\rho$ is the height of the hierarchy and $\mu$ is the fixed multiplier used to set the number of hidden nodes for concept $\gamma$ (see Section 3.2).

In a balanced decomposition, the number of hidden nodes for concepts reduces by a constant factor $\alpha$. With the size of smallest set of hidden nodes as $\delta$, the max layer of the dense connections is $\rho = \log_\alpha(\frac{d^0}{\delta})$. For any concept prediction $z^\gamma$, we need $d^\gamma = \frac{\mu d^0}{\alpha^l}$ weights at layer $l$ i.e., max number of weights for concept class prediction is $\mu d^0 \rho = \mu d^0 \log_\alpha(\frac{d^0}{\delta})$. The number of concept-concept connections can be calculated as $d^0(\frac{\mu}{\alpha} + \frac{\mu}{\alpha^2} + \cdots + \frac{\mu}{\alpha^\rho}) = \mu d^0 \frac{(1-\alpha^\rho)}{(1-\alpha)} \leq \mu d^0 \frac{1}{(1-\alpha)}$.

In order to predict a category variable $x_j \in \texttt{Children}(\gamma)$, $d^\gamma$ weights are necessary. Since any $\mu d^0 \geq d^\gamma = \mu \frac{d^0}{\alpha^l}$ at any level $l$ in a balanced decomposition, the total number of weights for category class prediction must be $\leq \mu d^0 N$.

## C  APPENDIX: TRAINING DETAILS

For MD-RN, MD-RN-FT and MD-IC training, we used the RMSProp optimizer with momentum in our multi-GPU distributed training scheme similar to Szegedy et al. (2016); Chollet (2016). The initial learning rate for this experiment was $0.1$ and was multiplied by $0.94$ every $2$ epochs. The epoch size is the same as the training set size with batch size $256$, momentum value $0.9$, and weight decay $0.0001$. The training data were augmented by random crop and horizontal flips. For proposed network learning, the weights corresponding to the concept outputs and the concept interconnections were learned for first $2$ epochs before optimizing those for the category variables. We did not use label smoothing for training the Inception v4 model. The training was continued until the CNN achieved the same or close accuracy for category classification as reported in the original paper/repository.

We have also trained a ResNet-50 with proposed multilayer dense connections MD-RN-SC and baseline flat dense connection BL-RN-SC from scatch on ImageNet 2012 training set. In our training from scratch, we applied stochastic gradient descent (SGD) optimizer with momentum value $0.9$ and initial learning rate $0.2$. The learning rate was decreased every $30$ epochs by a factor of $10$ (similar to Simonyan & Zisserman (2015); He et al. (2015) training). All other values of the hyperparameters remain the same as those of MD-RN training. Table 5 reports the accuracies of MD-RN-SC and BL-RN-SC.

Table 5: Accuracy comparison for learning full network from scratch. All accuracies are computed for the single crop top-1 setting. The proposed method achieves significantly higher $Acc_{\text{CON}}$, $Acc_{\text{COMB}}$ than the baseline.

| Method | $Acc_{\text{CAT}}$ | $Acc_{\text{CON}}$ | $Acc_{\text{COMB}}$ |
|---|---|---|---|
| BL-RN-SC | 75.24 | 69.87 | 59.7 |
| MD-RN-SC | 73.5 | 78.14 | **65.74** |

## D  APPENDIX: PERFORMANCE ANALYSIS OF MD-RN AND MD-IC ON IMAGENET

Figure 5(a) shows some example categories where the proposed MD-IC model generated different concept orders than those in the condensed ontology. Monitor and Space bar are children of Equipment and Implement concepts respectively in the condensed ontology. But, they contain many images of Desktop computer and Computer keyboards – both of which are children of the Device concept. As a result, 68% and 88% of Monitor and Space bar images respectively were classified as Artifact→ Instrumentality→ Device. Similarly, although it is grouped to Matter→ Solid concepts, 75% of the Mushroom images were predicted as Living thing which consists of Earthstar, Stinkhorn categories that are visually very similar to Mushroom images. Motor scooter was placed in Wheeled vehicle in our compressed hierarchy; but at test time, 94% of the time its images was predicted as Self propelled vehicle. In general, predictions from our proposed model concur more

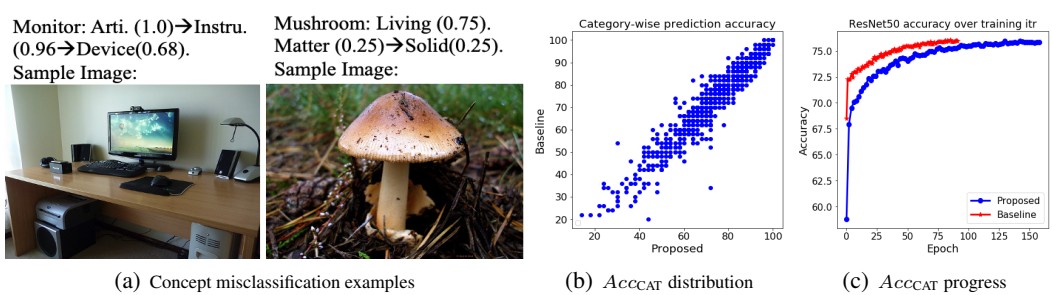

| (a) Concept misclassification examples | (b) $Acc_{\text{CAT}}$ distribution | (c) $Acc_{\text{CAT}}$ progress |
|---|---|---|

Figure 5: (a): Categories where the proposed method predicted a concept order different from the condensed hierarchy. (b) : Category-wise classification accuracy of the proposed method vs the baseline architecture (w/ ResNet50). (c): Progession of validation accuracy of the proposed CNN (blue) and baseline (red).

with the concept chains for descendants of Living thing than it does for those of Artifact concept.

Since different category classes are predicted at different layers of the proposed dense structure, it is rational to verify whether or not the category classification capability is impaired by the depth of layer. To test this, we have plotted in Figure 5(b) the category prediction accuracies for each of the $N = 1000$ classes of the baseline against those of the proposed CNN with ResNet50 backbone. The plot implies no clear effect of the prediction depths on the classification performances on different categories as they remain same or very close to those of the original architecture.

Due to the increased number of dense layers and additional sigmoid activations, it is perhaps natural to expect the proposed architecture to require more iteration to converge. As Figure 5(c) demonstrates, our model indeed takes more epochs(x-axis) to attain a category classification performance (y axis) similar to that of baseline built upon ResNet50.

## E    APPENDIX: ABLATION STUDIES

We have conducted ablation studies with model MD-RN. In our first ablation study, we model the dense layers based on the original uncompressed hierarchy instead of the condensed hierarchy. Adopting the original ontology in ResNet-50 increased the number of z variables to 806, depth of dense layers to 17 and model to 43M (compare with 40, 7, 30M resp. of compressed). Our attempts with different hyperparameters ($\lambda$, lr etc) achieved at most $\{Acc_{\text{CAT}}, Acc_{\text{CON}}, Acc_{\text{COMB}},$ mhP, mhR$\}$=$\{44.02, 1.12, 1.03, 60.05, 47.6\}$ at 30 epoch (compare $Acc_{\text{CAT}}$ with ours Fig 5(c). The training is perhaps smothered by 1) increased depth with many redundant single parent-child connections, and 2) imbalance in descendent distribution where smaller concept classes are underrepresented.

The second analysis is performed with varying the values of the balancing term $\lambda$ in the combined loss function (Section 3.2). As can be inferred from the network architecture for multiple dense connections and the loss functions, with $\lambda = 0$ the category-wise loss drive concept $z$ variables of only the ancestor concepts to 1. However, with $\lambda = 0$, the optimization will not force $z$ variables of the other non-ancestor concepts to 0. Consequently, concept and combined accuracy of a model trained with $\lambda = 0$ will be very low.

As Table 6 shows, we observe a very low $Acc_{\text{CON}}, Acc_{\text{COMB}}$ for MD-RN-$\lambda 0$ while achieving close $Acc_{\text{CAT}}$ to the published results. It is also apparent that a small $\lambda$ (= 2) favors improving $Acc_{\text{CAT}}$ while a relatively large $\lambda$ (=8) favors improving $Acc_{\text{CON}}$ compared to those of the MD-RN model reported on Table 1. This is expected behavior with the loss function we are optimizing.

Table 6: Performance of MD-RN with different $\lambda$ values. All accuracies are computed for the single crop top-1 setting.

| $\lambda$ | $Acc_{\text{CAT}}$ | $Acc_{\text{CON}}$ | $Acc_{\text{COMB}}$ | mhP | mhR |
|---|---|---|---|---|---|
| 0 | 76.04 | 1.86 | 1.57 | 33.35 | 45.41 |
| 2 | 76.04 | 78.57 | 66.66 | 91.78 | 93.81 |
| 8 | 75.42 | 81.74 | 70.08 | 94.13 | 94.08 |

## F    APPENDIX: PERFORMANCE ANALYSIS ON PASCAL VOC DATASET

We have excluded the classes {Person, Pottedplant} since these categories were underrepresented in the ImageNet12 dataset. For example, Person is represented by three rare subclasses Scuba diver, Ballplayer, Groom. On the other hand, the {Dog, Bird} categories were over represented in ImageNet12 and their subcategories were assigned to multiple concepts in our hierarchy. These classes could lead to inconsistency and therefore also excluded in evaluation. After removing the images that overlaps with multiple categories, we are left with 6941 images from 16 categories. For these images, we report the $Acc_{\text{CON}}$ in Table 7 ($Acc_{\text{CAT}}$ and $Acc_{\text{COMB}}$ cannot be computed as there is no one to one correspondence between categories of the two datasets).

The accuracy values for concept classification on VOC 2012 clearly indicate the superiority of the proposed architecture to generalize the knowledge it learned from ImageNet12 hierarchy of ancestor superclasses. All the CNN models resulted in weak performances of categories {Train, Car,

Table 7: Concept accuracy comparison on testing on PASCAL VOC 2012 trainval subset. The proposed method achieves significantly higher $Acc_{CON}$ than the baseline for both architectures.

| Method | cat | cow | horse | sheep | plane | boat | bike | mbike | bus | train | car | botl | chair | dtable | sofa | montr | avg |
|---|---|---|---|---|---|---|---|---|---|---|---|---|---|---|---|---|---|
| PrAggRN | 71.2 | 73.1 | 51.4 | 72.1 | 75.1 | 55.3 | 73.4 | 34.1 | 61.7 | 4.4 | 2.4 | 36 | 29.3 | 8.5 | 30.8 | 37.8 | 46.2 |
| BL-RN-FT | 57.2 | 53.9 | 43 | 53.8 | 72.5 | 50.2 | 65.7 | 35.3 | 56.8 | 1.5 | 1.6 | 21.6 | 17.9 | 1 | 13.1 | 30.7 | 38.06 |
| MD-RN-FT | 70.7 | 78 | 56.2 | 70.8 | 85.8 | 69.1 | 76.8 | 4.9 | 54.6 | 3.6 | 2.7 | 43.4 | 36.2 | 13.8 | 44.7 | 53 | **48.84** |
| PrAggIC | 69.8 | 80.7 | 59.8 | 80.7 | 80.3 | 59.7 | 75.4 | 30.1 | 64.6 | 4 | 2 | 38.9 | 31.8 | 9.5 | 36.4 | 33.3 | 48.32 |
| BL-IC | 64.9 | 63.2 | 47.2 | 60.5 | 80.3 | 45.9 | 45 | 2.9 | 71 | 1.9 | 2.1 | 25.3 | 19.3 | 3.1 | 24 | 32.3 | 40.3 |
| MD-IC | 74.6 | 84.1 | 60.9 | 80.7 | 86 | 63.1 | 77.5 | 8.1 | 65.8 | 3.8 | 1.7 | 46.4 | 39.4 | 15.9 | 40.9 | 43.3 | **50.38** |

Diningtable}. Such a performance can be attributed to equivocal ancestry within the condensed ontology. For the category Train, the proposed model with InceptionV4 architecture predicted Entity→ Artifact→ Instrumentality→ Container→ Wheeled vehicle → Self propelled vehicle for 56% of images whereas the compressed ontology assigns it the concept order : Entity→ Artifact→ Instrumentality→ Conveyance. It is important to note though that for the ambiguous categories, both the baseline and the proposed models performed poorly – i.e., the proposed method is not drawing any inequitable advantage due to the label ambiguity.

## G  APPENDIX: POSSIBLE APPLICATIONS AND EXTENSIONS

Our dense connectivity can also be extended to two-stage object detection algorithms (He et al., 2017; Lin et al., 2017) by replacing the final detection head with proposed mulitlayer connections. For SSD-type architectures (Liu et al., 2015; Lin et al., 2017), the dense computations can be carried out by $1\times1$ convolutions (or extended to $3\times3$) and added to the classification branches at each scale. The convolutional form should conceptually enable the semantic segmentation techniques (Chen et al., 2017; 2018; Zhao et al., 2016) to adopt our model as well.

Furthermore, rather than capturing the conceptual class lineage, the dense layers can be modeled after a different contextual relationship such as spatial or compositional consistency (via can-coexist-with (Hu et al., 2015) or is-part-of (Brust & Denzler, 2018; Wang et al., 2016) relations. These relationship graph, e.g., parsing graph (Tu et al., 2005), can be either precomputed or learned for the particular task at hand.

## H  APPENDIX: GROUPING CATEGORIES USING HDCNN TECHNIQUE

We also attempted to group the ImageNet 2012 categories following the strategy suggested by the HDCNN paper, Section 4 of (Yan et al., 2015). For this purpose we exploited the ResNet-50 classifier trained on 93% training examples of ImageNet 2012 (i.e., the initial classifier for HG-RN). The remaining 7% of the training set was used to compute 100 disjoint clusters of categories. While some of the animal and vegetable/fruit categories were correctly grouped into the same clusters, many of the clusters comprises semantically very different classes. We are listing a few of these clusters below. Yan et al. (2015) further created some overlapping groups for their algorithm. It is not clear how to assign concept class labels to these undersegmented clusters of categories and how they can be predicted using a classifier. That is why we cannot compare our method with HDCNN.

```
Cluster 1 : {n02509815 lesser-panda, n02346627 porcupine,
n01872401 echidna, n02172182 dung-beetle, n02319095 sea-urchin,
n03717622 manhole-cover, n04033901 quill, n03733281 maze}.

Cluster 2 : {n04086273 revolver, n02804414 bassinet, n03954731
plane, n03995372 power-drill, n03483316 hand-blower, n03868863
oxygen-mask, n03759954 microphone, n02841315 binoculars, n02749479
assault-rifle, n04090263 rifle, n03602883 joystick, n04485082
tripod, n04517823 vacuum, n04554684 washer, n03724870 mask,
n04069434 reflex-camera, n03424325 gasmask}.

Cluster 3: {n02389026 sorrel, n02403003 ox, n02437312
Arabian-camel, n03538406 horse-cart, n03868242 oxcart, n04417672
thatch}.
```

```
Cluster 4: {n03447447 gondola, n02951358 canoe, n04204347
shopping-cart, n03933933 pier, n04311004 steel-arch-bridge,
n04366367 suspension-bridge, n04532670 viaduct, n03160309 dam,
n03873416 paddle}.
```

# I APPENDIX: ROBUSTNESS OF MD-RN TRAINED ON AWA2

In order to test the robustness of the proposed MD-RN network trained on AwA2 dataset, we evaluate its performance on ImageNet 12 images with overlapping categories. There are 16 AwA2 categories that are also present in ImageNet 12 dataset. We applied our MD-RN trained on AwA2 on 800 ImageNet 12 validation images and achieved category, concept and combined accuracies $\{Acc_{\text{CAT}}, Acc_{\text{CON}}, Acc_{\text{COMB}}\} = \{89.87, 89.37, 85.75\}$. For comparison, we also applied the flat baseline BL-RN on these images and achieved $\{Acc_{\text{CAT}}, Acc_{\text{CON}}, Acc_{\text{COMB}}\} = \{88.87, 78.12, 74.12\}$. This experiment shows that the proposed dense layers in MD-RN learned on AwA2 can generalize better than the single layer in BL-RN. Note that, the convolutional feature layer of both MD-RN and BL-RN have not changed because only dense layers of these two networks were trained (see Section 4.2). Figure 6 shows a few examples of output from the proposed network on ImageNet 12 validation images.

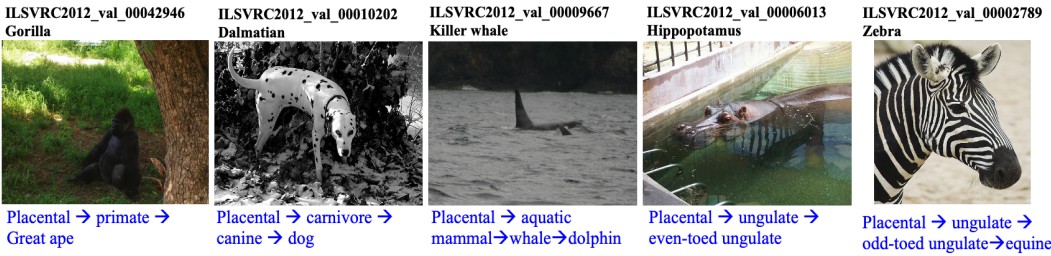

Figure 6: Qualitative performance of MD-RN trained from AwA2 datasets on images from ImageNet 12 validation set. The image ids and MD-RN category predictions are listed on top of each image whereas the concept predictions are displayed below.

