# OpenReview forum: "Multilayer Dense Connections for Hierarchical Concept Classification"
_ICLR.cc/2021/Conference — Reject_

### Official Review · AnonReviewer4 · 2020-10-24
**solution looking for a problem?**

**Rating:** 3
**Confidence:** 4

**Review:**

The paper proposes a method to learn concept classes along with its concept superclasses. The proposed method relies on an ontology which they heuristically re-organize by essentially pruning nodes that have few descendants and large semantic overlap. The network proposed to model the ontology essentially just consists of a learned multiplicative gate at each level of the ontology with a standard xent loss over concepts and regularizing term that indicates if the category is an ancestor concept. The experimental results claim gains over some baselines, e.g., combined acc. of 69.05 vs chosen baseline 66.15 on ImageNet12 for ResNet-50 features at a cost of between 18.2% increase in parameters.

Overall, while some of the empirical results seem competitive, I am most concerned with the weak foundations of the motivation of the setup. The work reads like a proposed solution that's trying to look for a problem / motivation and as a result struggles to find its footing in explaining modeling choices & results.

* The paper uses as motivation that many networks use a softmax head over semantic categories at the leaves of an ontology and claims this is therefore why models using such networks do not learn that say English Setter is a dog. This is a shallow argument for incorporating concept hierarchies since such models clearly would still not be learning deeply what the concept of a dog is, only encoding weak priors introduced by the ontology, an external knowledge base from the network. The connection to learning relationships like "is-a" relations don't ultimately fall out from the proposed method, instead you just get a list of likelihoods that correspond to superclasses that contributed to a concept prediction—the model is not learning the relation, just the co-presence of these superclasses.
* The argument that works like Deng et al (2012, 2014), which for example propose label relation priors like HEX graphs, only either predict fine-grained labels or superclasses exclusively, but not both simultaneously, is another example of where this paper falls short in its problem setup. This work doesn't answer the question of _why_ one would even want to predict both simultaneously well. If we believe that a superclass is unlikely present, why would we still predict the child classes? Even if there are reasonable arguments for this, they are absent in this paper.
* The approach to creating the "compressed concept hierarchy" largely felt like a description of what was done, again, rather than why. Unless I missed it, I also expected a baseline for ablation that doesn't use the "compressed" hierarchy, but just uses it as-is.
* It's a bit strange to me why a MSE loss is used for the indicator of whether a concept is an ancestor. Why use an unbounded error in L2, even if (or especially if) you are squeezing through a sigmoid? What is the intuition?
* I would be curious to know if the improvement in results that we see in Table 1 are just due to increased model capacity (params/compute), i.e. how does it compare to the Deng et al baselines. The comparison discussed are only made with respect to the ResNet-50 & Inception-V4 backbones.

I will also note that the paper could improve on its clarity in writing. As an example, from Figure 2, it's unclear what exactly changed from the LHS and RHS, how, and why it's meaningful; and in Figure 3, it's not obvious without work how the input, output and z-term relate.

---

> ### Author Response · Authors · 2020-11-12
> **Thanks very much for your feedback -- is it possible the benefits of the proposed method were overlooked?**
>
> Is it not correct to state that combined accuracy improvement is 70.06 vs 66.15 (or 70.45 vs 66.15 in revised version) with an 18.2% increase in parameters? Fine tuned version MD-RN-FT does not increase the number of parameters.
>
> Addressing the concerns in the same order as pointed out:
>
> * We agree the statements regarding learning representation of concept classes were sloppy and will rewrite them in the revised version. We request the reviewer to pls see the response to Rev2 to avoid repetition.
>
>   In addition, the value of variable $z^\gamma$ prediction of a child concept (e.g., primate) is conditioned upon the state of $z^{parent(\gamma})$ of its parent (e.g., mammal) due to the multiplicative gates (Eqn 2). That is, the detection of any concept depends upon the detection of its parents. The proposed method therefore captures the hierarchical dependence among concepts and categories, not merely the co-occurence of them.
>
> * One could ask how one would confidently believe a concept is absent during the test phase. The work of Deng et al.12 incorporates this decision within their algorithm and attempts to predict either a category or an ancestor superclass that maximizes the reward. In doing so, their algorithm seems to lose category classification accuracy with respect to the initial classifier, i.e., it fails to recognize the categories of a significant percentage of examples that the initial classifier (ResNet50 and Inception V4) can already provide us (Section 5.1 Evaluation in original submission, Section 4.2 in revised version). Similarly, the approach of Deng et al.14 fails to achieve the category accuracy of its backbone (AlexNet, see paper).
>
>   On the other hand, the proposed method produces both the category and concept predictions, in x and z vectors respectively, so that the user or a particular application can threshold these values to determine which subset of superclasses may be present in the image and discard the rest. The difference between Deng et al. 12, 14 is that we let the subsequent user/application to decide which concepts/categories may be present (in a highly accurate prediction) as opposed to embedding this decision within the algorithm.
>
>   Hypothetically speaking, we wonder whether the reviewer thinks one should attempt to predict both the category and sequence of concepts **or**  predict either of these exclusively in an ideal scenario. Could the reviewer image the benefit of both the category and concept classification in applications as those indicated in the last paragraph of the introduction and perhaps beyond?
>
> * An ablation study with uncompressed hierarchy is provided in Appendix E first paragraph. In short, the presence of many redundant parent-child relations and an imbalance in the descendant distribution impedes the network with an uncompressed hierarchy to achieve good performance.
>
> * We do not believe the MSE loss is crucial for learning the concept classes, using a binary cross-entropy for each concept may work as well.  We are not sure why choice of this loss function is critical for the design and overall performance of our architecture.
>
> * We do not think we understood the reviewer's comment on this. Our version of Deng et. al. 12 uses  ResNet-50 and Inception V4 as initial classifiers (Section 4 Baselines in original submission, Section 4.1 Baseline in the revised version).  The output of these CNNs are passed to the optimization scheme of Deng 12. On the other hand, the proposed model replaces the dense layers of ResNet-50 and Inception with the multilayer dense connections. Could the reviewer pls elaborate a little to help us understand? The original version of Deng et.al. 12 uses SVM classifier on features derived from SIFT and assigns at most 40% images of Imagenet 1K to leaf categories.

---

### Official Review · AnonReviewer1 · 2020-10-27
**Presentation of the papers needs improvement**

**Rating:** 5
**Confidence:** 3

**Review:**

Summary:
This paper proposes a novel module on top of ConvNet, multi-layer dense connectivity, for learning hierarchical concepts in image classification.

Pros:

This paper proposes to use the label hierarchy (with ancestor concepts from a label) instead of the label itself to learn the image recognition system. To achieve this, it has made two major contributions:
1. Building label hierarchy with a simplified set of categories, to remove the redundant and meaningless categories
2. With the constructed label hierarchy, this paper proposes a dense connectivity module to leverage the label hierarchy to model category abstractions over high-level visual embedding, on top of commonly used convolutional neural networks.

With the proposed techniques, this paper builds up its recognition system using two standard deep ConvNets and achieved strong results on large-scale image recognition benchmarks.

Cons:

1. In general, the paper is not very well written for a few reasons: A) The motivation of the proposed method over previous methods is not clear (intro paragraph#2). B) Section 3.1 is very hard to follow. C) Some notations in Section 3.2 seems unnecessary and there are things being used before it is formally defined.

2. The design of this dense connectivity module in Section 3.2 seems quite arbitrary, there is no good explanation on why we need to use the z to multiply the output x and h.

3. Experiments of naturally adversarial examples are not motivated earlier in the introduction. It's quite hard for me to understand why using a label hierarchy would improve this task.


Detailed Comments:

1. Paragraph#2 in Intro: why training a neural network as multinomial/softmax logistic regression from images to labels can not acquire a comprehensive knowledge about the input entity? For instance, in some of the prior works (e.g. Hu et. al. 2018), they learn models to simultaneously classify categories on a predefined label hierarchy, including both abstracted classes such as "Dog" and concrete class such as the "English Setter".
2. It seems that from Section 3 on, it uses the term "Category" to stand for the leaf concept (most specific) and the term "Concept" as the shorthand of ` "Ancestor Concept". It would be better to mention this explicitly to avoid confusion.
3. Example in Figure 2 is not very clear and hard to follow. It might be better to simplify the figure by using a smaller hierarchy as an example. Also, it would be good to have a paragraph in section 3.1 to describe what in the right figure has been modified using the concrete examples of Figure 2.
4. Equation 1, why do we need a \psi activation function which is linear? What it means by linear, is there an additional linear weight in \psi besides v?
5. Why are we using an MSE for the concept classifiers? I assume we can use binary cross-entropy for them?


Minor:
* The aspect ratio of Figures 2 and 3 need to be adjusted. It is hard to recognize text and symbols on the stretched figures.
* Notation \hat{h} in the text is bolded but the ones in the equation (1) is not bolded
* A recent work that also leverages hierarchical information in the label text to learn visual concept embeddings, which is closely related to the topic of this paper: Learning to Represent Image and Text with Denotation Graph. EMNLP 2020

---

> ### Author Response · Authors · 2020-11-12
> **Thanks a lot for your efforts to understand our paper.**
>
> * We agree the statements regarding learning representation of concept classes were sloppy and will rewrite them in the revised version. We request the reviewer to pls see the response to Rev2 to avoid repetition. We will also modify the presentation based on the reviewer’s suggestions.
>
> * As Rev4 pointed out, the values of $z^\gamma$ is exploited for multiplicative gating so that any of its children concept $z^{child(\gamma)}$ or category $x_j$ is detected only if the concept $\gamma$ is detected. The lines below Eqn (2) points out how $z^\gamma$ acts in either an excitatory or inhibitory roles for its children.
>
> * The $\psi$ in Eqn (1) is actually an identity function, sorry to confuse you with an informal notation.
>
> * We are applying our method to the AwA2 dataset in order to compare its performance with that of Hu et al 16. The results will be included in the revised version to be uploaded soon.
>
> * We agree with the reviewer that using a binary cross-entropy for each concept may also work as well, MSE loss is not essential for learning the $z$ variables.

---

### Official Review · AnonReviewer3 · 2020-10-29
**Great topic, interesting but not very general model, weak baselines.**

**Rating:** 5
**Confidence:** 3

**Review:**

The authors consider how to capture the semantic relationship among categories of a classifier. It is an important problem and has many potential applications. For example, the predicted concept chain can help people understand the performance of the classifier, the coarse-grained concepts are beneficial to the few-shot learning of new categories, and etc.
The authors incorporate WordNet as their ontology and build their neural classifier based on it. Such a tree-structured dense-connected neural architecture is not very common in the current deep learning domain. The network is bound to external ontology, so when the ontology updates, the network has to be re-built. In my opinion, the design seems not very general. Maybe the authors could consider representing restrictions among concepts in the vector space.
In the experiments, the authors used only two baselines, one is a flat single-layer classifier, the other is a work of 2012. The baselines seem too weak to demonstrate the superiority of the proposed model. The results of more recent works are necessary, even though these works "use a separate technique/tool for modeling the conceptual relations", as the authors claimed.

To sum up, the pros of this paper include:
- a valuable research topic
- a fancy model
- clear experimental details

The cons include:
- the model is not very general
- baselines are too weak
- the aspect ratio and resolution of figures seem improper

---

> ### Author Response · Authors · 2020-11-12
> **Thanks very much for your comments.**
>
> **Re: Generality** In terms of the hierarchy, it would change only if the domain is different, i.e., medical images as opposed to natural images or if we wish to capture a different interrelation among classes, e.g., spatial organization for scene parsing. If one aims to model the semantic ancestry among categories, the hierarchy would not change for a task that employs classification (e.g., image recognition, object detection or semantic segmentation). This is because WordNet hierarchy is a general, sufficiently inclusive ontology for objects of interest in the physical world, which is why multiple works (e.g., Deng et al, 12, 14; Hwang & Sigal 14, Hu et al. 16) rely on it.
>
> In terms of the network architecture, it would change (at least the dense layers) for a different set of target labels for any CNN, wouldn’t it? Could the reviewer pls elaborate, preferably with examples, why our proposed method is less general than other relevant studies?
>
> **Additional baselines:**  Just in case Rev3 missed it, we already compare with one method, namely the  *improved* version of Deng et.al. 2012, that uses a separate tool for computing concepts.  As mentioned in Section 4 Baseline (Section 4.1 Baseline in revised version), our modified version of Deng et.al. 12 uses CNNs ( HG-RN uses ResNet and HG-IC uses Inception) as opposed to the classical methods employed in the original work.
>
> We are applying our method to the AwA2 dataset in order to compare its performance with that of Hu et al 16. The results will be included in the revised version to be uploaded soon.

---

### Official Review · AnonReviewer2 · 2020-11-02
**insufficient experimental justification, missing citations**

**Rating:** 2
**Confidence:** 5

**Review:**

This paper designs a multilayer connection structure for neural networks, such that the connection architecture supports implementation of a hierarchical classification scheme within these layers.  It applies this design to the task of hierarchical classification on ImageNet.  Experiments compare results with those of Deng et al. (2012), as well as baseline flat classification models.

The paper motivates the proposed approach via broad claims about what networks understand, but does not provide sufficient analysis or experimental evidence to justify these claims.  For example:

"In particular, when an existing CNN correctly identifies an image of an English Setter, the network itself does not learn that it is an instance of a dog, or more precisely, a hunting dog which is also a domestic animal and above all, a living thing"

Assuming it is trained on example images of all of these categories, how do we know that the CNN does not learn shared representations that implicitly reflect such organization of the concept space?  The paper does not employ any techniques to probe the learned representations of CNNs; without such analysis, the sweeping statements about what CNNs do or do not learn are mere speculation.

On a technical level, the design of the proposed dense classification layers appears to be quite ad-hoc.  It is not clear why a special design intermixing concept prediction nodes with hidden nodes is desirable or necessary.  How does this compare, both conceptually and experimentally, to a branching hierarchy of subnetworks?  The scheme of HD-CNN (Yan et al., 2015) is similar to the latter, but is not represented in experimental comparisons.

In fact, experiments appear to lack comparison to any recent published methods on hierarchical classification.  Deng et al. (2012) is the only prior publication that serves as a reference point.  This is far from a sufficient baseline as surely there has been other work on hierarchical classification in the past 8 years.  For example, a highly relevant work that this paper fails to even cite or discuss is:

M. Nickel and D. Kiela. Poincare Embeddings for Learning Hierarchical Representations. NeurIPS, 2017.

Together, the unsubstantiated motivating claims, ad-hoc design of questionable merit, limited experimental comparison, and missing citations to highly relevant recent work suggest that this paper is not of sufficient quality for publication.

---

> ### Author Response · Authors · 2020-11-12
> **Statements may be misleading but the findings/solution have merit?**
>
> We thank the reviewer for the feedback.
>
> **Re: claims on learned representation:** We agree with the reviewers that the statements regarding what representations CNNs learn were sloppy. We will rephrase these statements in the next revision and upload it soon -- we will incorporate the suggestions emerging from the discussion as well.
>
> Is it correct to state that classification performance is one measure of strength of a representation? If the existing CNNs, e.g., ResNet and Inception, learn a representation as informative for conception classification as it is for categories, it is logical to expect the classification performances to be similar with the same structure of multinomial regression. This is exactly what we tested with the single dense layer flat baseline predicting category and concept classes independently. The fact that the concept accuracy of BL-RN, BL-RN-FT, BL-IC  is worse than category accuracy by 7~14% (compare Acc_CON and Acc_CAT of these networks on Table 1) suggests that the representations learned by these CNNs are less informative for concept classification than they are for category classification. Would the reviewers agree to this logical deduction?
>
> **Comparison with HDCNN:** In Section 2 & Appendix H, we explained why our method cannot be compared with HDCNN of (Yan et al. 2015). In short, the HDCNN algorithm does not use the WordNet hierarchy -- it clusters the categories to compute its own superclasses which generally does not lead to semantically meaningful concepts. HDCNN further utilizes overlapping clusters for their algorithm which renders predictions of concept classes to be very difficult, if not impossible.
>
> Moreover, HDCNN was designed for a hierarchy with only 1 level (i.e., height=1). It does not seem straightforward (if not extremely difficult) to extend the connectivity between coarse and fine label components, and the final output, of HDCNN to a multilevel hierarchy (e.g., height=7) for concept and category prediction. Could the reviewer pls shed some light on it, and also perhaps on the complexity and training of the resulting model?
>
> There have been only a handful of studies that exploit the concept labels for classification within these past 8 years -- we are currently applying our method to the AwA2 dataset in the hopes to compare it with Hu et al. CVPR16. We would like to point out here that our improved version of (Deng et al. 2012) uses ResNet and Inception V4 as initial classifiers.
>
> **Re: Hyperbolic embedding:** The work of hyperbolic embedding maps the inputs to an embedding space where it was shown that similar categories reside nearby. Nickel et al. NIPS17 use words, not images -- although a more recent work by Khrulkov et al. CVPR20 proposes hyperbolic embeddings for images. The image hyperbolic embedding of Khrulkov et al.  groups together images from semantically similar categories -- it is not clear how to retrieve the concept classes from the embedded points, the authors did not propose a method for it. The category classification may be obtained via a protypical representation which is suitable for few-shot learning and there will be questions whether or not it is fair to use it for standard image classification. Perhaps the reviewer could suggest how to compare the performance experimentally with ours?
>
> One can imagine our method  projects an image to an embedding captured in the concept prediction vector $\mathbf{z}$. The difference is our embedding space is predefined whereas  Khrulkov et al. learns it. We intentionally left out this line of work to circumvent confusion, and due to space limitation, but will cite in the updated version.

---

### Author Response · Authors · 2020-11-16
**Revision with modified text and additional results**

We thank the reviewers again for their comments on our paper and wish to direct them to the uploaded revision. A brief summary of the modifications/additions to the revision provided below.

**Writeup**:  The statements on learning feature representation have been removed and/or rephrased in the revised manuscript. We wish to reiterate that sloppy description does not invalidate the findings/improvements demonstrated in the paper.

The text, captions and figures have also been modified for clarification in accordance with reviewer suggestions. We will continue to improve the exposition based on suggestions from all reviewers till the final revision deadline.

For the convenience, the changes in the revision have been highlighted in purple.

**Experiments**: We have added a new experiment that applies our proposed method on AwA2 dataset in order to compare with Hu et.al. CVPR16 paper (as suggested by Rev1 and 3). Recall Hu et.al.16 predicts ancestor concepts for AwA2 and follows a hierarchy similar to that of ImageNet12 dataset. We have followed the experimental setup of Hu et.al. as closely as possible and report accuracy with the same evaluation quantities.

Although we use a more capable CNN backbone than that of Hu et.al. (ResNet vs AlexNet), the significantly better performance of the proposed technique for all three types of classes (category, concept, attribute) implies that with the current popular network architectures, our multilayer dense connections may be adequate for a correct hierarchical concept classification -- i.e., a separate tool such as RNN may not be necessary for this task. We also verify experimentally that the performance improvement does not stem solely from the more sophisticated backbone by comparing against flat baselines comprising ResNet-50. Our method achieved substantially higher accuracy for concept and attribute classification that the single layer flat baseline strategy.

In addition, we updated the accuracies of the MD-RN-FT with slightly better values that we attained before the discussion period.

We kindly request the reviewers to peruse the revised version and provide feedback on whether the modified text/additional material addresses their concerns or there are still issues that we can address or materials we can supply.

---

### Decision · Program_Chairs · 2021-01-07
**Final Decision**

**Decision:**

Reject

**Comment:**

All reviewers recommended rejection after considering the rebuttal from the authors. The main weaknesses of the submission include poorly motivated claims and designs, and insufficient experimental comparisons. The AC did not find sufficient grounds to overturn the reviewers' consensus recommendation.